# Identifying and Modeling Resonance-Related Fluctuations on the Experimental Characteristic Impedance for PCB and On-Chip Transmission Lines

Yojanes Rodríguez-Velásquez [1], Reydezel Torres-Torres [2]* and Roberto Murphy-Arteaga [2,*]

1   INTEL Guadalajara, Zapopan 45017, Mexico; yojand16@gmail.com
2   Electronics Department, Instituto Nacional de Astrofísica, Óptica y Electrónica, Tonantzintla,
    Puebla 72840, Mexico; reydezel@inaoep.mx
*   Correspondence: rmurphy@ieee.org; Tel.: +52-222-736-9044

**Abstract:** It is well known that the fluctuations in experimentally obtained characteristic impedance versus frequency curves are associated with resonances originated by standing waves bouncing back and forth between the transitions at the transmission line terminations. In fact, microwave engineers are aware of the difficulty to completely remove the parasitic effect of these transitions, which makes obtaining smooth and physically expected frequency-dependent curves for the characteristic impedance a tough task. Here, we point out for the first time that these curves exhibit additional fluctuations within the microwave range due to standing waves taking place within the transition itself. Experimental verification of this fact was carried out by extracting this fundamental parameter from measurements performed on on-chip and printed circuit board (PCB) lines using probe pad adapters and coaxial connectors. We demonstrate that the lumped circuit approach to represent the transitions lacks validity when the additional fluctuations due to the connectors become apparent, and we propose a new model including transmission line effects within the transition.

**Keywords:** characteristic impedance; *S*-parameter measurements; interconnect transitions

## 1. Introduction

The fundamental parameters for the representation of the electrical properties of uniform transmission lines (TLs) guiding signals in single mode are the propagation constant ($\gamma$) and the characteristic impedance ($Z_c$) [1]. In this regard, whereas the determination of $\gamma$ is straightforward, solving eigenvalue equations involving measurements of lines varying in length, obtaining $Z_c$ is cumbersome [2,3]. This is because this latter parameter is strongly affected by the return loss of the TLs [4]. In fact, even after applying the most advanced de-embedding methods to the measurements, significant fluctuations in the $Z_c$ curves are observed at microwave frequencies from experimental data [5–8]. These fluctuations are associated with the interaction of standing waves with the imperfect terminations of the TLs, which are associated with the signal launchers that serve as the interface between the lines and the test equipment. Specifically, the fluctuations occur at resonant frequencies where half the wavelength of the signal equals the physical length of the line. Furthermore, we show here for the first time—to the best of our knowledge—that additional fluctuations become apparent due to standing waves taking place within the line's transition itself, since these transitions present a non-zero physical length.

To contextualize the reader, in this paper several popular methods to obtain $Z_c$ are applied to illustrate the effect of the transitions on the experimentally obtained curves. In this regard, in the traditional method, single line measurements are used, but resonances of large magnitude affect the extraction [3,9], even after removing inductive and capacitive parasitics associated with the line terminations [10]. Alternatively, a de-embedding procedure involving measurements performed on two lines varying in length can be applied [11].

Nevertheless, this method assumes that the parasitics at the terminations can be effectively represented using a single shunt admittance, which provides acceptable results for lines on-chip but fails when the transition requires a representation considering series and shunt effects. This is the case of lines on packaging technology and also on-chip when the wavelength of the signal is comparable to the physical length of the interconnect.

Another approach uses $\gamma$ data to obtain $Z_c$, for instance by considering the relationship between these parameters and the resistance, inductance, conductance and capacitance (*RLGC*) per unit length elements in the model of a line operating in the transverse electromagnetic mode [12]. The advantage of this method is that relative permittivity and loss tangent data can be used to infer the complex curves for $Z_c$ from $\gamma$ [13]. Nevertheless, this method can only be applied in a straightforward manner when the capacitance and conductance parameters in the *RLGC* model are related to well identified dielectric effects, which hampers its application to lines on lossy substrates (e.g., metal-insulator-semiconductor microstrip lines). Furthermore, as frequency increases, the wavelength of the propagating signals may become comparable to the physical length, not only of the uniform transmission line section but also to that of the signal launchers terminating the lines. In this case, extracting the characteristic impedance of the uniform section of the transmission line becomes more complicated since the measured data correspond to the concatenation of three transmission lines: the uniform section of line embedded between two transmission lines corresponding to the signal launchers.

In this paper, to contribute to the representation of microwave transmission lines using either pad-arrays or connectors as measurement interfaces, a new model including the distributed effects associated with the signal launchers is proposed. To wit, since the corresponding impact is of great relevance on the measured return loss of the lines, the limitation of common characteristic impedance determination methods when ignoring this additional effect at microwave frequencies is fully demonstrated.

## 2. Materials and Methods

Accurate knowledge of the complex $Z_c$ is necessary for TL characterization, as well as for calibration procedures. Thus, determining this parameter from *S*-parameter measurements is desirable. In this regard, in an early paper about this topic, the transmission parameters obtained from an *S*-to-*ABCD* parameter transformation of a TL exhibiting a length *l* are assumed to be given by [5]:

$$\mathbf{T}_h = \begin{bmatrix} \cosh(\gamma l) & Z_c \sinh(\gamma l) \\ \sinh(\gamma l)/Z_c & \cosh(\gamma l) \end{bmatrix} \tag{1}$$

Thus, $Z_c$ can be straightforwardly obtained as:

$$Z_c = \sqrt{\mathbf{T}_h\,[1,2]/\mathbf{T}_h\,[2,1]} \tag{2}$$

where the numbers in brackets indicate the element position in the $\mathbf{T}_h$ matrix. Unfortunately, the transitions that interface the line with the test equipment at the measurement plane after calibration introduce effects that are typically represented by lumped capacitive (*C*) and inductive (*L*) parasitics. Figure 1a shows a schematic illustrating a way to account for these parasitics, which allows the following expression to be written for the associated transmission parameters:

$$\mathbf{T}_{LhL} \approx \begin{bmatrix} 1 & j\omega L \\ j\omega C & 1 \end{bmatrix} \mathbf{T}_h \begin{bmatrix} 1 & j\omega L \\ j\omega C & 1 \end{bmatrix} \tag{3}$$

Notice that applying (2) to (3) no longer yields $Z_c$, but curves that include fluctuations or glitches introduced by the effect of *C* and *L*. These glitches are observed in the Re($Z_c$) curves in the form shown in Figure 1b when either the inductive or capacitive effects are dominant. In fact, the periodicity and magnitude of these glitches is dependent on the

length of the TL as illustrated in Figure 1c [9]. Observe, however, that the fluctuations occur around the expected value for $Z_c$, which suggests that they can be easily removed.

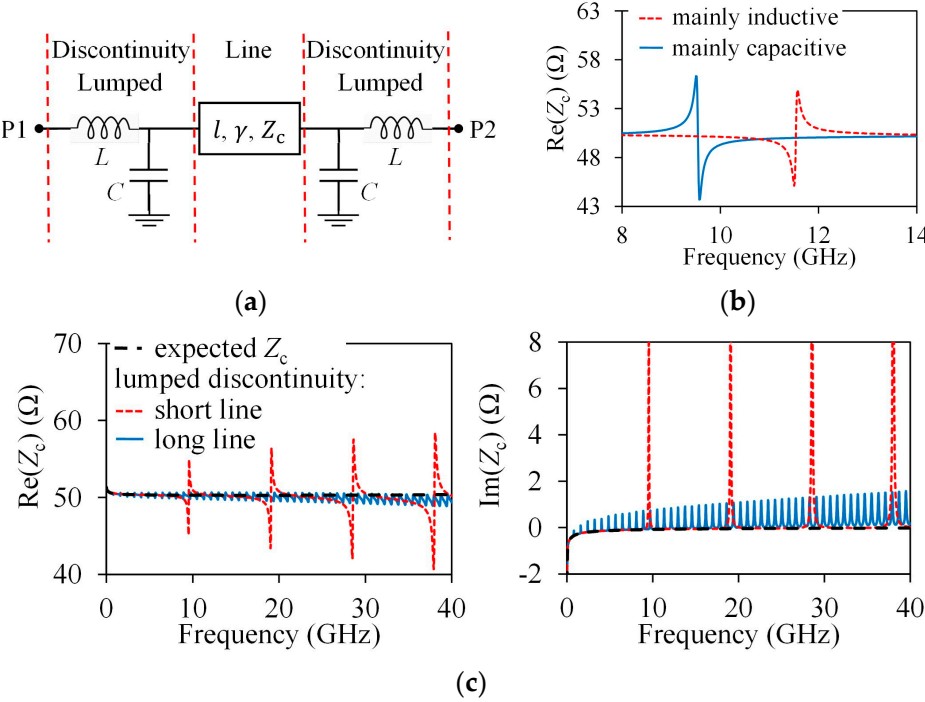

(a)

(b)

(c)

**Figure 1.** Representation of the effect of lumped transition parasitics in $Z_c$ curves: (**a**) schematic of a uniform TL embedded between the transitions, (**b**) curves showing the fluctuations in $Z_c$ when either the inductive or capacitive effects are dominant and (**c**) $Z_c$ calculated from (2) when the parasitics are ignored.

For microwaves, however, the distributed nature of the transitions is accentuated, and it is then evident that the model for a measured line should represent part of the transition as a TL exhibiting a propagation constant $\gamma_D$ and a characteristic impedance $Z_D$. This model is shown in Figure 2a, whereas the associated transmission parameters are:

$$\mathbf{T}_{DLhLD} = \mathbf{T}_D \mathbf{T}_{LhL} \mathbf{T}_D \tag{4}$$

where

$$\mathbf{T}_D = \begin{bmatrix} \cos h(\gamma_D l_D) & Z_D \sin h(\gamma_D l_D) \\ \sin h(\gamma_D l_D)/Z_D & \cos h(\gamma_D l_D) \end{bmatrix} \tag{5}$$

In this case, in addition to the effect of $C$ and $L$, the distributed effect associated with the transition discontinuity introduces oscillations that further complicate the determination of the frequency-dependent $Z_c$ (see Figure 2b).

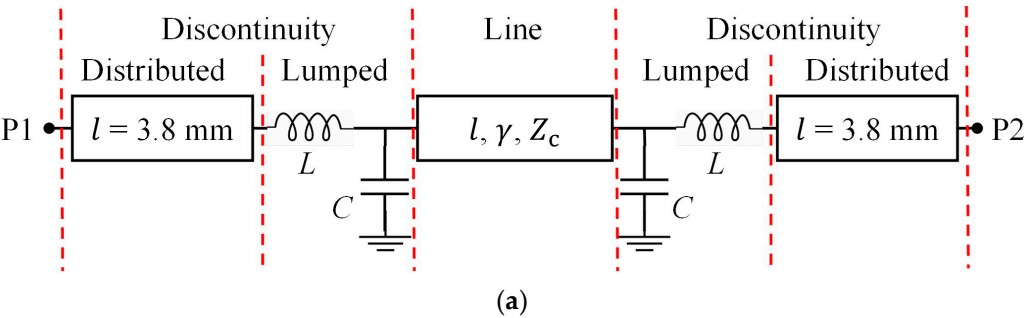

(a)

**Figure 2.** *Cont.*

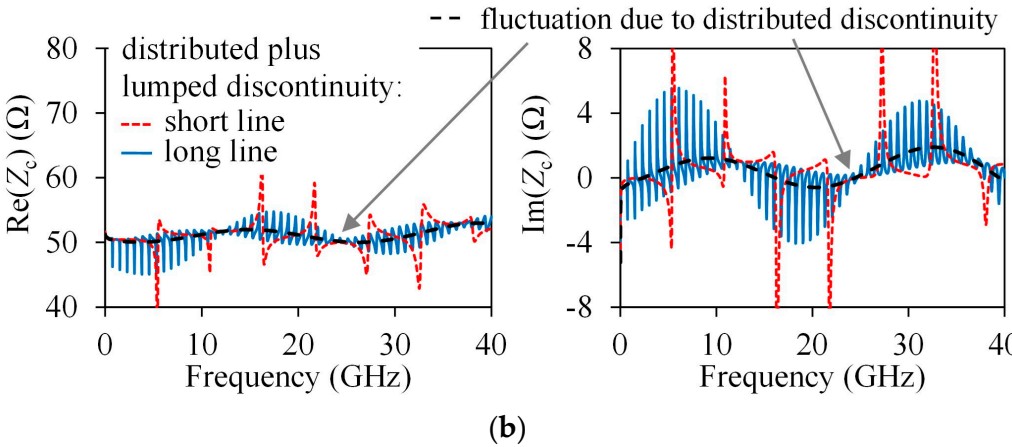

**Figure 2.** Illustrations depicting the effect of parasitics in $Z_c$ curves when the transition includes effects that are required to be represented using a distributed model: (**a**) schematic showing a uniform TL embedded between the transitions and (**b**) $Z_c$ calculated from (2) when the parasitics are ignored.

## 3. Results

### *A    Lines on-chip*

TLs were fabricated on a CMOS process; their cross section is shown in Figure 3a. These lines present lengths of $l = 1380$ μm, 2450 μm and 4600 μm and include a metal patterned ground shield separating a distance $h = 0.8$ μm from the signal trace [14,15]. Moreover, the lines are terminated with ground–signal–ground (GSG) pad arrays, which allow for the measurement of *S*-parameters using coplanar RF probes with a 150 μm pitch. In addition, "open" and "short" dummy structures were included to allow the application of the open-short de-embedding procedure.

### *B    Lines on PCB terminated with probe pad adapters*

Microstrip lines with the cross section and dimensions shown in Figure 3b were formed on a PCB laminate with nominal permittivity and loss tangent of 2.2 and 0.0019, respectively [16]. The lengths of the lines are $l = 12.7$ mm and 101.6 mm, and present a design $Z_c \approx 51$ Ω. The measurements were performed using GSG probes with a 150 μm pitch.

### *C    Lines on PCB terminated with coaxial connectors*

Figure 3c shows the cross section of the microstrip lines prototyped on a PCB, which exhibits nominal permittivity and loss tangent of 3 and 0.002, respectively. The lines were designed to present $Z_c \approx 72$ Ω and are terminated with 40 GHz general precision connectors (GPCs) with a 2.92 mm interface [17,18]. In this case, the VNA setup was calibrated up to the coaxial interface [19].

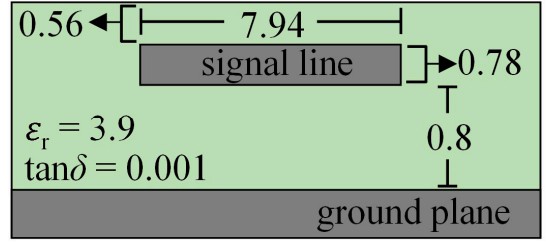

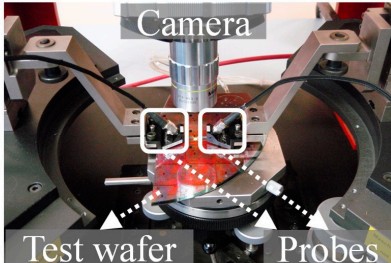

(**a**)

**Figure 3.** *Cont.*

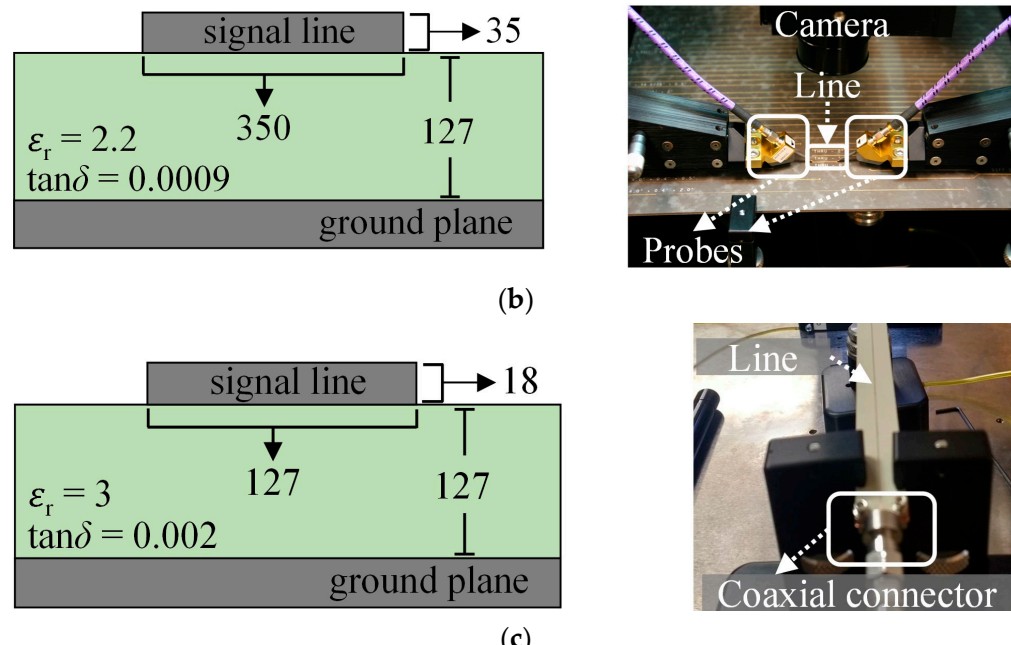

**Figure 3.** Sketches of the cross sections exhibited by the prototyped lines accompanied by photographs of the corresponding measurement setup, which has been previously calibrated: (**a**) on-chip lines terminated with probing pads, (**b**) PCB lines terminated with probing pads and (**c**) PCB lines terminated with coaxial connectors. All dimensions are expressed in microns (microns).

## 4. Discussion

For microstrip lines manufactured on PCB, it is expected that the $|Z_c|$ curve does not vary considerably with frequency since the losses are relatively small. Conversely, for microstrip lines made on silicon, the films of the signal traces are thin enough to have per unit length (PUL) resistances of the order of $k\Omega/m$. This high resistance causes the characteristic impedance to have a strong variation over a wide range of frequencies [20]. For the on-chip lines, the $Z_c$ obtained using (2) for the different line lengths is shown in Figure 4. In addition to the fluctuations due to the transitions, $Z_c$ exhibits large discrepancies depending on the line length, which is unexpected. On the other hand, using the open–short method [21], the fluctuations due to discontinuities are smoothed; however, an unexpected roll-down of the $Z_c$ is noticeable as frequency increases. This variation is originated because the open and short structures consider that the transition between the pads and the lines is abrupt. In contrast, using the line–line method [11], a flat $Z_c$ is achieved from 10 to 50 GHz, which can be considered closer to the expected value. In this latter case, the extraction method relies on the fact that the transition can be represented by means of a lumped shunt admittance, which is valid provided that the pad array can be considered relatively small, a condition fulfilled for on-chip interconnects but not for PCB lines.

On PCB lines, the PUL resistance is considerably smaller, but the associated length might be so large that a significant number of fluctuations on $Z_c$ can be observed within a range of a few tens of gigahertz. In Figure 5, $Z_c$ extracted using (2) is shown for two lines of different length, as well as that extracted from $\gamma$ [13]. As can be seen, the fluctuations due to reflections within the line length have a greater magnitude for the shorter line but occur at a higher rate on the longer line. Observe in Figure 5 that, in reference to Figure 1b, the form of the curves evidences a dominant inductive effect associated with the transition. In contrast, $Z_c$ obtained from $\gamma$ is smooth, so it can be taken as a good approximation for lines with these characteristics. The disadvantage of this procedure is that it requires previous knowledge of the frequency-dependent complex permittivity. Otherwise, a constant effective permittivity and loss tangent can be assumed, but causality in the representation of the line might be compromised.

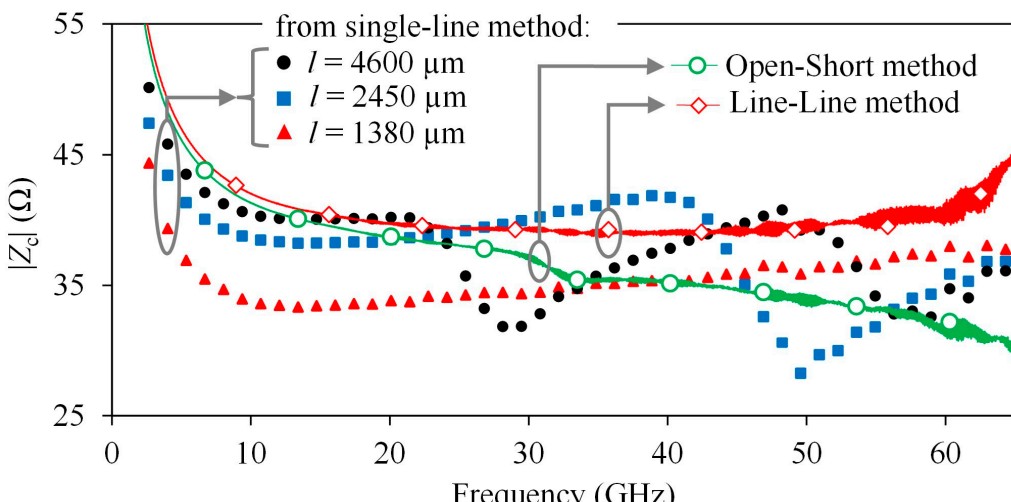

**Figure 4.** $|Z_c|$ obtained for interconnects on-chip from direct measurements, line–line method and open-short method.

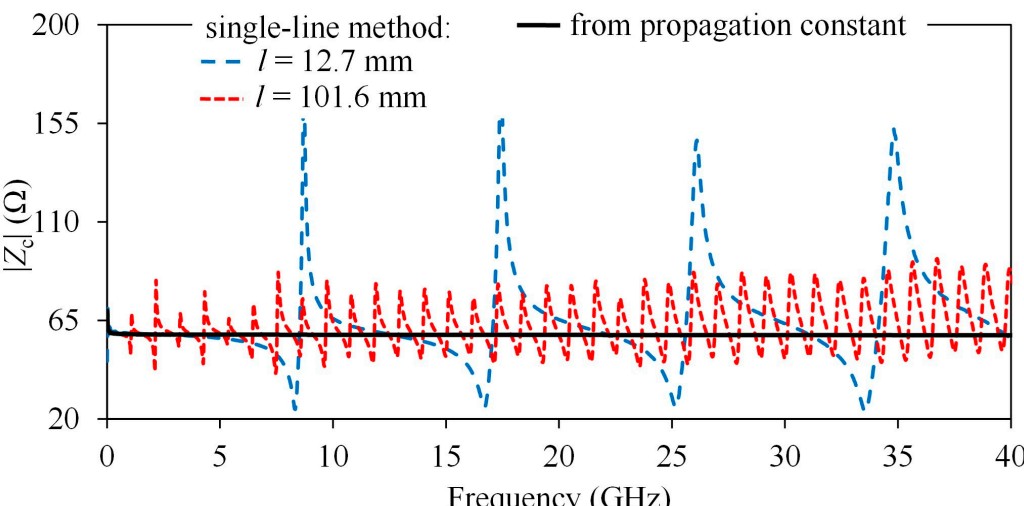

**Figure 5.** $|Z_c|$ for PCB lines interfaced with probe pad adapters; curves obtained from direct measurements applying (2) to the two available line lengths and from propagation constant are shown.

For the third case, the PCB lines terminated with coaxial connectors exhibit an additional effect that considerably complicates the extraction of $Z_c$ from measured S-parameters. Due to the electrical length of the coaxial-to-microstrip transitions, a distributed model for the discontinuity needs to be considered, so the interconnect plus the transitions needs to be modeled with (4). In Figure 6, $Z_c$ obtained using (2) as well as from $\gamma$ is shown. Unlike the case where probes were used to measure, fluctuations not only occur with a periodicity related to the length of the lines, but $Z_c$ presents an additional fluctuation of lower periodicity, which is related to reflections along the length of the coaxial connectors. Although these reflections are present in any type of discontinuity, they become more noticeable in this case for the considered frequency ranges. Therefore, when the length of the transition is considerably long, it makes it even more difficult to determine the actual frequency dependence of $Z_c$ from experimental data. Interestingly, when comparing the curves in Figure 6 with the conceptual plots in Figure 1b, it can be observed that at frequencies below 15 GHz, the transition exhibits a dominant capacitive behavior, which becomes inductive at higher frequencies. This is indicative of the importance of implementing a broadband model for this type of microwave transition measurement.

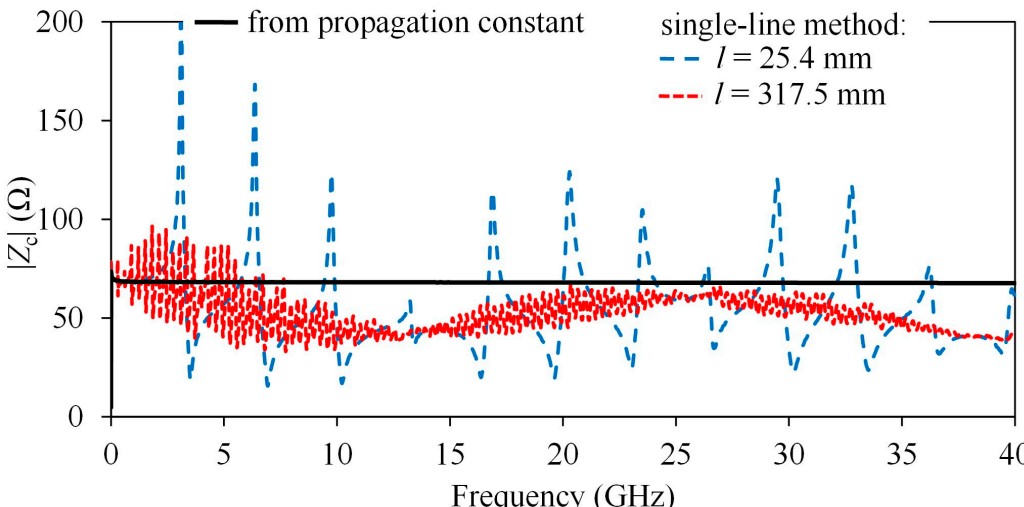

**Figure 6.** $|Z_c|$ obtained for interconnects on PCB with coaxial connectors from direct measurements (Equation (2)), and also from propagation constant data using the method in [13].

The mentioned fluctuations are transferred to the *RLGC* parameters shown in Figure 7. Here, the lines labeled "from propagation constant" refer to (2) considering constant values for permittivity and loss tangent, which are not true in practice. Observe that the fluctuations are so large that the expected frequency variation of the curves, predicted by the data obtained when $Z_c$ is obtained from $\gamma$, is barely noticeable.

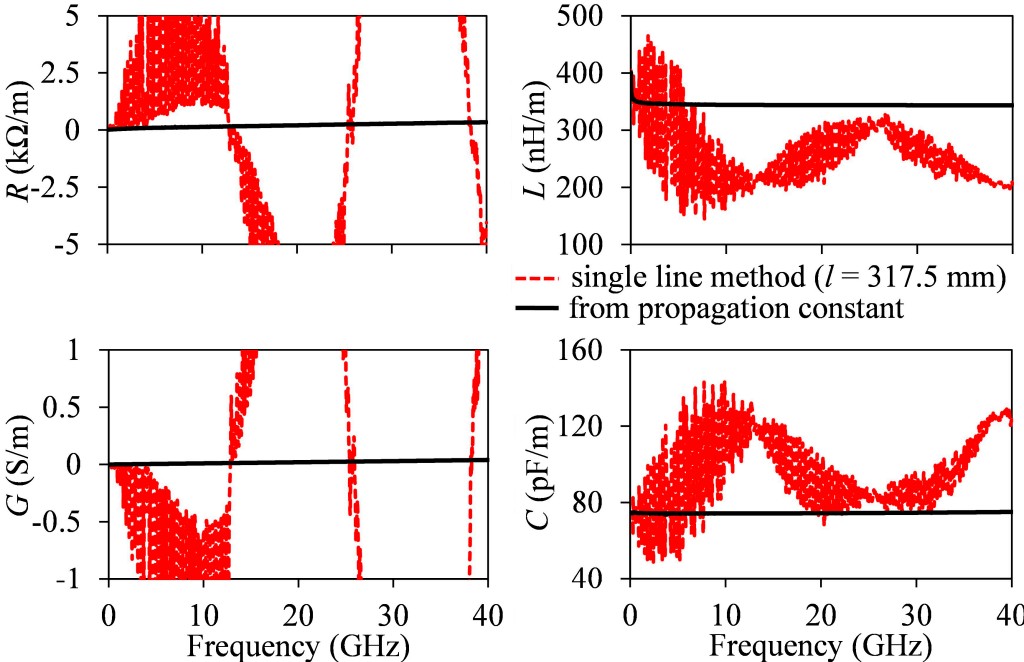

**Figure 7.** *RLGC* parameters obtained for interconnects on PCB terminated with coaxial connectors. These parameters consider $Z_c$ extracted from direct measurements (Equation (2)), and also from propagation constant data using the method in [13].

In order to account for the launch transition, a model was derived considering it as a distributed coaxial line, derived using ADS. This model is shown in Figure 8, and a comparison of this model with experimental data is shown in Figure 9. The good agreement of the model with the experimental data indicates that the distributed representation of the coaxial connectors allows the reflection effects to be characterized in relatively large

transitions. As can be seen in Figure 9, using a lumped *LC* model for the coaxial connectors produces good results up to 10 GHz, but for higher frequencies, the representation is not good enough. The lumped model represents the termination only as a reflective structure; therefore, using this approximation, the impedance of the connector might be acceptably represented, but it is assumed that no standing waves would occur within the connectors themselves. Nonetheless, as frequency increases, the wavelength of the propagating signals is reduced and may become comparable with the physical length of the connectors. In this case, the LC circuit does not capture the distributed effect of the connectors, and therefore, the model fails to represent the multi-transmission line structure composed by the microstrip lines embedded between coaxial connectors that in reality behave as coaxial transmission lines at high frequencies.

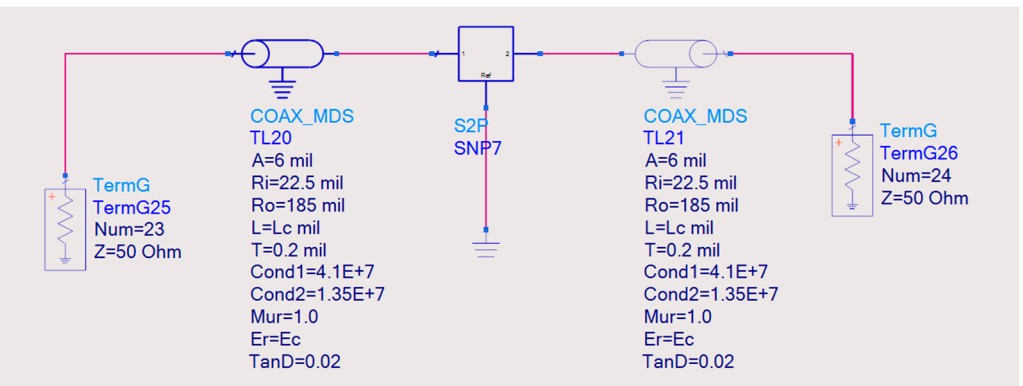

**Figure 8.** A model for the transmission line including the transitions, represented here by distributed coaxial lines. The coaxial lines were of length Lc = 3.8 mm and relative permittivity Ec = 2.5. Homogeneous transmission line *S*-parameters, represented by the S2P box, were obtained from extracted $\gamma$, $Z_c$ approximated from $\gamma$, and $l$ = 25.4 mm.

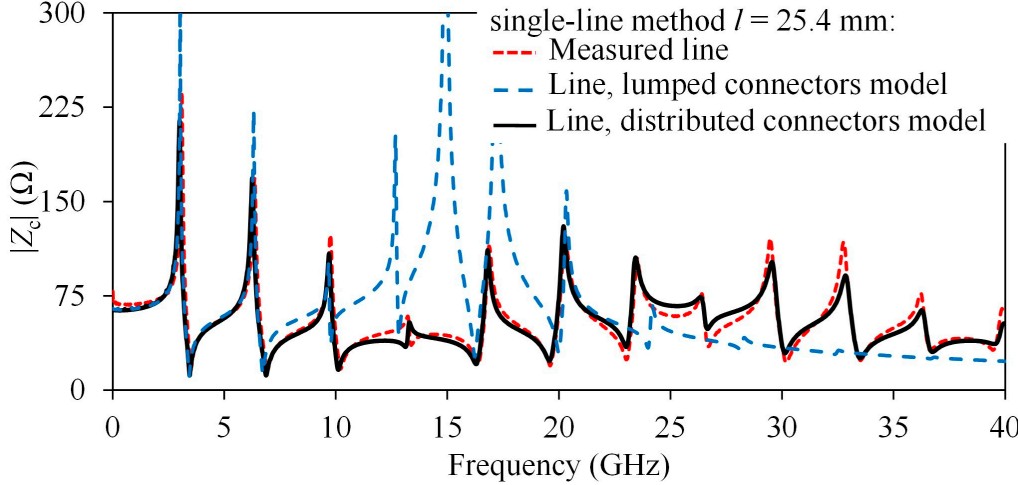

**Figure 9.** Effective impedance measurement-model comparison for a PCB line terminated with a coaxial connector. A lumped model for the coaxial connectors was added for comparison purposes with inductance $L_{COAX}$ = 0.995 nH and capacitance $C_{COAX}$ = 0.38 pF.

## 5. Conclusions

The objective of this study was to analyze and explain additional resonances evident when measuring transmission lines. These resonances are attributed to the connectors used to probe the line and are produced by standing waves along the line due to the influence of the line in conjunction with the probing platforms and the connectors employed to probe the lines. In order to accurately determine characteristic impedance of the line, these

resonances have to be identified and quantified. With this aim in mind, three different structures were manufactured, probed and analyzed to determine the corresponding effect on experimentally derived characteristic impedance data. The analyses show that short lines, for instance on-chip, are less impacted by this undesired effect when compared to lines on PCB. In fact, the typical line–line algorithm, assuming that the transitions are simply modeled using a shunt admittance, provides acceptable results up to some tens of gigahertz. Nonetheless, for long lines, such as those on PCB, the fluctuation effect is considerable and is further worsened when using transitions that also exhibit a noticeable distributed nature within the measurement range. This later effect was identified and is presented in this paper using a transmission line distributed model.

Clearly, some approximations were made in this study, the most important being the fact that the transitions on either side of the transmission line were taken as identical. This assumption might lead to additional complications by not considering the different impedance values that can arise from the reflected waves at either end. We can expect a difference in the transitions due to possible mismatch during the manufacturing process, the positioning of the probes and the soldering of the connector, among others.

We are working on refining the method, and in future endeavors we aim at repeating the experiments with different connectors, lengths and widths of lines, as well as substrates, in order to garner more and deeper detail on the effect while avoiding the approximations considered thus far.

**Author Contributions:** Conceptualization, Y.R.-V., R.T.-T., R.M.-A.; methodology, Y.R.-V., R.T.-T., R.M.-A.; validation, Y.R.-V., R.T.-T., R.M.-A.; formal analysis, Y.R.-V., R.T.-T., R.M.-A.; investigation, Y.R.-V., R.T.-T., R.M.-A.; writing—original draft preparation, Y.R.-V.; writing—review and editing, R.T.-T., R.M.-A.; project administration, R.T.-T., R.M.-A.; funding acquisition, R.T.-T., R.M.-A. All authors have read and agreed to the published version of the manuscript.

**Funding:** This research was funded by the Mexican National Science and Technology Council (Conacyt) grant numbers 285199 and 288875 and scholarship number 719285.

**Data Availability Statement:** Data is available upon request.

**Conflicts of Interest:** The authors declare no conflict of interest.

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
