# Peer review of "Identifying and Modeling Resonance-Related Fluctuations on the Experimental Characteristic Impedance for PCB and On-Chip Transmission Lines"

_electronics, doi:10.3390/electronics12132994_

Round 1

Reviewer 1 Report

This paper deals with studies on the impedance characteristics of transmission lines made on PCB substrates. Although the subject is well-known and widely reported in the literature, the authors observed additional fluctuations in these characteristics.  

As highlighted in the abstract this was done for the first time (line 17). This is where my first comment and concern arises and relates to the literature review.  In my opinion, there is a lack of analysis of recent sources on the subject. Most of the references are from 10 or more years ago.

In addition, there is a lack of any detailed description of the proposed computational model (tool, simulation assumptions, accuracy and error analysis.

Overall, I find the article interesting and the results may have practical application and be of interest to engineers and, with some additions, may be recommended for publication.

Generally, the article was written in a comprehensible language, although it should be  checked for English language and editing.

 Please check the text in terms of linguistic, grammar and punctuation.

In my opinion, a scientific text should be written in the passive voice (e.g. is shown in Fig.x instead of Fig. shows ..., etc). I found this in some places.

Author Response

Reviewer 1:

This paper deals with studies on the impedance characteristics of transmission lines made on PCB substrates. Although the subject is well-known and widely reported in the literature, the authors observed additional fluctuations in these characteristics.  

Thank you very much for your valuable comments!  No doubt these will serve to improve the quality of our report.

As highlighted in the abstract this was done for the first time (line 17). This is where my first comment and concern arises and relates to the literature review.  In my opinion, there is a lack of analysis of recent sources on the subject. Most of the references are from 10 or more years ago.

We considered the most relevant references to our work, but based on your comment we have studied some more recent ones and have included them in the reference list.  These are now [6]-[8] in the manuscript:

6.   H. Bai, F. Gan, N. Zhang, and Y. Li, Characteristic Impedance Analysis of Transmission Lines Considering Frequency Characteristics, in Proceedings of the 2018 3rd Advanced Information Technology, Electronic and Automation Control Conference (IAEAC 2018), Chongqing, China, pp. 1804-1808, Oct. 2018.

7.   C.C. Huang, Determination of Characteristic Impedance of Planar Transmission Lines in Lossy/Dispersive Substrates by Using Series Resistor with Frequency-Dependent Inductance, in IEEE Trans. on Microw. Theory and Tech., Vol. 68, No. 10, pp. 4229-4235, Oct. 2020.

8.   M. Panduro, J.A. Reynoso-Hernández, A method for determining the characteristic impedance of transmission lines embedded in transitions, in Int. J. Electron. Commun. (AEÜ), 66, pp. 185-188, 2012.

In addition, there is a lack of any detailed description of the proposed computational model (tool, simulation assumptions, accuracy and error analysis.

The model was derived and verified using ADS (Advanced Design System, by PathWave Design, a division of Keysight Technologies).  In ADS one can input experimental data in order to optimize parameter values.

Overall, I find the article interesting and the results may have practical application and be of interest to engineers and, with some additions, may be recommended for publication.

Thank you very much!  We also believe so.

Comments on the Quality of English Language

Generally, the article was written in a comprehensible language, although it should be  checked for English language and editing.

 Please check the text in terms of linguistic, grammar and punctuation.

In my opinion, a scientific text should be written in the passive voice (e.g. is shown in Fig.x instead of Fig. shows ..., etc). I found this in some places.

We have proofread the paper carefully, and we believe it is now free of errors.

Reviewer 2 Report

This paper deals with microwave measurements of the characteristic impedance of guiding structures that exhibit resonance fluctuations caused by transitions to terminations.

1) The problem addressed in the paper is not new and is relatively simple, for the whole structure is assumed to be longitudinally symmetric, with identical input and output transitions/terminations.

2) A lot of theoretical and experimental work has already been published on the topic, even considering asymmetrical terminations. The Reference list is very short, and the authors’ claim «…for the first time…» in L233 (Conclusions) should be avoided.

3) The content is not very significant, nor very sound.

Frequently, a scientific paper on microwaves is well stuffed with math equations (this is not mandatory but is usual). This paper has five, trivial, and a key one is wrong.

The transfer matrix, or ABCD matrix, in (3) is calculated from the concatenation of three reciprocal networks. The matrices on the left and right of T_h are written as if they are equal. Unfortunately, not only they are unequal, but, also, they should have determinant equal to 1, which is not observed. In fact, the element A in the left-matrix must be A = (1-wLwC), and the element D in the right-matrix must be D = (1-wLwC). The importance of the neglected term LCw^2 rapidly increases with the frequency, and I wonder if the authors’ conclusions would be the same if this term were accounted for.

Another content-related question is: In Fig. 1a) and Fig. 2a) the networks to the left and to the right of the device under test are symmetrical of each other. I understand the authors’ choice (to keep the input and output ports of the whole structure indiscernible), but reality is what it is.

Does the measurement setup ensure that such a theoretical assumption is fulfilled?

If for any reason (and there could be many) the global two-port structure contains different left and right transitions, it will be characterized by 1 propagation constant, but 2 characteristic impedances (with different absolute values) will be required, not one. I saw no reference to this real problem.

Another question: Which values (and why) have been assigned to the lumped parameters L and C for modeling purposes? Are they frequency invariant from 0 to 40 GHz?

 4) Comments on units.

The lengths and sizes in Fig. 3 lack units, millimeters, micrometers, mils? Note that Fig. 8 displays mils.

Kilo Ohm is not written as KOhm. Capital K means Kelvin. Kilo is non-capitalized.

Author Response

Reviewer 2:

This paper deals with microwave measurements of the characteristic impedance of guiding structures that exhibit resonance fluctuations caused by transitions to terminations.

Thank you very much for your valuable comments!  No doubt these will serve to improve the quality of our report.

1) The problem addressed in the paper is not new and is relatively simple, for the whole structure is assumed to be longitudinally symmetric, with identical input and output transitions/terminations.

The problem might seem simple, but it has been the focus of many studies throughout the years.  Herein we are presenting a small advancement to the field of knowledge by considering the effect the connectors have on the measured data.

2) A lot of theoretical and experimental work has already been published on the topic, even considering asymmetrical terminations. The Reference list is very short, and the authorsclaim «…for the first time…» in L233 (Conclusions) should be avoided.

We have added three more recent references to the article, and one of these deals precisely with set-up transitions, without identifying the effects we have described in this paper.

3) The content is not very significant, nor very sound.

Frequently, a scientific paper on microwaves is well stuffed with math equations (this is not mandatory but is usual). This paper has five, trivial, and a key one is wrong.

We have kept the mathematics to a minimum since the effect object of the paper is modeled electrically, and not mathematically.  Indeed, a further study of the problem might lead to a mathematical formalism to explain it, but that will not necessarily be simple.

The transfer matrix, or ABCD matrix, in (3) is calculated from the concatenation of three reciprocal networks. The matrices on the left and right of T_h are written as if they are equal. Unfortunately, not only they are unequal, but, also, they should have determinant equal to 1, which is not observed. In fact, the element A in the left-matrix must be A = (1-wLwC), and the element D in the right-matrix must be D = (1-wLwC). The importance of the neglected term LCw^2 rapidly increases with the frequency, and I wonder if the authorsconclusions would be the same if this term were accounted for.

The determinant for the left block is:

The reviewer is right that the left and right blocks might not be equal, but it is common practice to take them as such.

We are taking impedance and capacitance values from our reference [10] in the revised version of the manuscript.  From the table in page 178, we get L~20.2 pH, C~10.2 fF.  These are considered frequency invariant, and we can neglect the  term up to the frequency we are reporting is , and thus the left and right matrices are equal.

Another content-related question is: In Fig. 1a) and Fig. 2a) the networks to the left and to the right of the device under test are symmetrical of each other. I understand the authorschoice (to keep the input and output ports of the whole structure indiscernible), but reality is what it is.

Does the measurement setup ensure that such a theoretical assumption is fulfilled?

If for any reason (and there could be many) the global two-port structure contains different left and right transitions, it will be characterized by 1 propagation constant, but 2 characteristic impedances (with different absolute values) will be required, not one. I saw no reference to this real problem.

The problem was solved by interchanging ports and performing the measurements again. Since the difference was minimum, and it can be attributed to experimental errors, we concluded that the input and output transitions are equivalent, if not identical.

Another question: Which values (and why) have been assigned to the lumped parameters L and C for modeling purposes? Are they frequency invariant from 0 to 40 GHz?

 4) Comments on units.

The lengths and sizes in Fig. 3 lack units, millimeters, micrometers, mils? Note that Fig. 8 displays mils.

Thank you for your observation, we have included the units in the caption, all are in microns.

Kilo Ohm is not written as KOhm. Capital K means Kelvin. Kilo is non-capitalized.

We apologize for the mistake; this has been corrected in Fig. 7.

Reviewer 3 Report

Reviewer Comments

Manuscript Title: Identifying and Modeling Resonance-Related Fluctuations on the Experimental Characteristic Impedance for PCB and On-Chip Transmission Lines

Manuscript Number: electronics-2474947

The manuscript under review is devoted to studying the fluctuations associated with resonances originated by standing waves bouncing back and forth between the transitions at the transmission line terminations. The authors present experimental verification of this fact and demonstrate additional fluctuations within the microwave range due to standing waves taking place within the transition itself. In addition, the authors propose a new model including transmission line effects within the transition.

The manuscript contains new and significant. The abstract clearly and accurately describes the content of the article. The literature review part contains distinct and rich references. The paper is nicely written and can be accepted but first, it should be improved. I have these comments:

1- The paper contains a lot of typos and sentence structure, please revise it.

2- Some parameters and abbreviations (for example RLGC) are not defined in the text, please check.

3- The size of the majority of the images is too small, it must be enlarged.

4- The results of previous studies are not well explained in the introduction par it will be better if the authors add new recent references and compare their results with other works.

5- It is better if Figures 3 and 4 will be moved to the end of Part C of section 3 (Results section).

6- The conclusion does not correctly present the results found, please try to reformulate it

Finally, I recommend the paper for publication after resolving the comments.

1- The paper contains a lot of typos and sentence structure, please revise it.

Author Response

Reviewer 3:

Manuscript Title: Identifying and Modeling Resonance-Related Fluctuations on the Experimental Characteristic Impedance for PCB and On-Chip Transmission Lines

Manuscript Number: electronics-2474947

The manuscript under review is devoted to studying the fluctuations associated with resonances originated by standing waves bouncing back and forth between the transitions at the transmission line terminations. The authors present experimental verification of this fact and demonstrate additional fluctuations within the microwave range due to standing waves taking place within the transition itself. In addition, the authors propose a new model including transmission line effects within the transition.

The manuscript contains new and significant. The abstract clearly and accurately describes the content of the article. The literature review part contains distinct and rich references. The paper is nicely written and can be accepted but first, it should be improved. I have these comments:

Thank you very much for your valuable comments!  No doubt these will serve to improve the quality of our report.

1- The paper contains a lot of typos and sentence structure, please revise it.

We have proofread the paper and corrected the mistakes, thanks for the observation.

2- Some parameters and abbreviations (for example RLGC) are not defined in the text, please check.

They are defined in the revised version of the paper.

3- The size of the majority of the images is too small, it must be enlarged.

We have enlarged all the figures.

4- The results of previous studies are not well explained in the introduction par it will be better if the authors add new recent references and compare their results with other works.

We have included more recent references in the revised version of the paper, these are:

  1. 6. Bai, F. Gan, N. Zhang, and Y. Li, “Characteristic Impedance Analysis of Transmission Lines Considering Frequency Characteristics”, in Proceedings of the 2018 3rd Advanced Information Technology, Electronic and Automation Control Conference (IAEAC 2018), Chongqing, China, pp. 1804-1808, Oct. 2018.
  2. 7. C. Huang, “Determination of Characteristic Impedance of Planar Transmission Lines in Lossy/Dispersive Substrates by Using Series Resistor with Frequency-Dependent Inductance”, in IEEE Trans. on Microw. Theory and Tech., Vol. 68, No. 10, pp. 4229-4235, Oct. 2020.
  3. 8. Panduro, J.A. Reynoso-Hernández, “A method for determining the characteristic impedance of transmission lines embedded in transitions”, in Int. J. Electron. Commun. (AEÜ), 66, pp. 185-188, 2012.

5- It is better if Figures 3 and 4 will be moved to the end of Part C of section 3 (Results section).

We have moved both figures to the end of Part C, thanks for the suggestion!

6- The conclusion does not correctly present the results found, please try to reformulate it

Thanks for your observation!  We have rewritten the Conclusion section to better explain what the results of this work amounted to.

Finally, I recommend the paper for publication after resolving the comments.

 Thank you very much!  We also believe it is a good report of our work!

Comments on the Quality of English Language

1- The paper contains a lot of typos and sentence structure, please revise it.

We have proofread the paper carefully, and we believe it is now free of errors.

Reviewer 4 Report

(1)The largest problem in this manuscript is that the explanation and analysis of the results from the mechanism and physical principle are missing. For instance, on page 8, lines 215 - 217, "As can be seen in Fig. 9, using a lumped LC model for the coaxial connectors produces good results up to 10 GHz, but for higher frequencies, the representation is not good enough." The authors need to explain the reasons from the mechanism and physical principle why using a lumped LC model for the coaxial connectors produces good results only up to 10 GHz.

(2)The unit in Fig. 3 is missing and needs to be added into the figure.

There are some errors and typos in English in this manuscript. For instance, in the caption of Fig. 9, "Efective impedance measurement-model comparison for a PCB line terminated with a coaxial connector" needs to be corrected as "Effective impedance measurement-model comparison for a PCB line terminated with a coaxial connector".

Author Response

Reviewer 4:

(1)The largest problem in this manuscript is that the explanation and analysis of the results from the mechanism and physical principle are missing. For instance, on page 8, lines 215 - 217, "As can be seen in Fig. 9, using a lumped LC model for the coaxial connectors produces good results up to 10 GHz, but for higher frequencies, the representation is not good enough." The authors need to explain the reasons from the mechanism and physical principle why using a lumped LC model for the coaxial connectors produces good results only up to 10 GHz.

What we mean is the following:

The lumped model represents the termination only as a reflective structure; therefore, using this approximation the impedance of the connector might be acceptably represented, but it is assumed that no standing waves would occur within the connectors themselves. Nonetheless, as frequency increases, the wavelength of the propagating signals is reduced and may become comparable with the physical length of the connectors. In this case, the LC circuit does not capture the distributed effect of the connectors, and therefore, the model fails to represent the multi-transmission line structure composed by the microstrip lines embedded between coaxial connectors that in reality behave as coaxial transmission lines at high frequencies.

We have included this paragraph in the text to clarify the idea.

(2)The unit in Fig. 3 is missing and needs to be added into the figure.

Thank you for your observation, we have included the units in the caption, all are in microns.

Comments on the Quality of English Language

There are some errors and typos in English in this manuscript. For instance, in the caption of Fig. 9, "Efective impedance measurement-model comparison for a PCB line terminated with a coaxial connector" needs to be corrected as "Effective impedance measurement-model comparison for a PCB line terminated with a coaxial connector".

We have proofread the paper and corrected the mistakes, thanks for the observation.

Round 2

Reviewer 1 Report

Thank you for including my comments in the final version of the article. In my opinion, in its current form, it can be published in Electronics.

Author Response

Comments and Suggestions for Authors

Thank you for including my comments in the final version of the article. In my opinion, in its current form, it can be published in Electronics.

Thank you very much for your valuable comments and suggestions.  These helped us improve the quality of our report substantially.

Reviewer 2 Report

The authors' response to my comments was clear. This R1 version is better than the original.

In any case I believe that the Conclusion section should include two additional paragraphs. One, outlining the approximations/limitations of the work; refering in particular the issue of asymmetrical ends (and the complication arising from the existence of 2 characteristic impedances, for the forward and for the backward waves). And, another paragraph mentioning future work.

Minor edditing.

Author Response

Reviewer 2:

Comments and Suggestions for Authors

The authors' response to my comments was clear. This R1 version is better than the original.

In any case I believe that the Conclusion section should include two additional paragraphs. One, outlining the approximations/limitations of the work; refering in particular the issue of asymmetrical ends (and the complication arising from the existence of 2 characteristic impedances, for the forward and for the backward waves). And, another paragraph mentioning future work.

We would like to thank the reviewer again for his valuable and insightful comments, which undoubtedly helped us improve the quality of our report.  Regarding this comment, we have added two more paragraphs at the end of the Conclusions Section that read:

Clearly, some approximations were made in this study, the most important being the fact that the transitions on either side of the transmission line were taken as identical. This assumption might lead to additional complications by not considering the different impedance values that can arise from the reflected waves at either end.  We can expect a difference in the transitions due to possible mismatch during the manufacturing process, the positioning of the probes, and the soldering of the connector, among others.

We are working on refining the method, and in future endeavors we aim at repeating the experiments with different connectors, lengths and widths of lines, as well as substrates, in order to garner more and deeper detail on the effect, while avoiding the approximations consider thus far.

Reviewer 3 Report

The authors improve the article and respond to all questions and remarks. I suggest accepting the paper in its current form.

Author Response

Reviewer 3:

Manuscript Title: Identifying and Modeling Resonance-Related Fluctuations on the Experimental Characteristic Impedance for PCB and On-Chip Transmission Lines

Manuscript Number: electronics-2474947

Comments and Suggestions for Authors

The authors improve the article and respond to all questions and remarks. I suggest accepting the paper in its current form.

We would like to thank the reviewer for all the valuable comments and suggestions that translated into a substantially better report of our work; we appreciate the effort an time taken to peruse our manuscript.
